# Wisdom of the Machines: Exploring Collective Intelligence in LLM Crowds

**Yashar Talebirad**
University of Alberta
Edmonton, Canada
talebira@ualberta.ca

**Ali Parsaee**
University of Alberta
Edmonton, Canada
parsaee@ualberta.ca

**Vishwajeet Ohal**
University of Alberta
Edmonton, Canada
ohal@ualberta.ca

**Amirhossein Nadiri**
York University
Toronto, Canada
anadiri@yorku.ca

**Csongor Szepesvári**
University of Alberta
Edmonton, Canada
csongor@ualberta.ca

**Yash Mouje**
University of Alberta
Edmonton, Canada
mouje@ualberta.ca

**Eden Redman**
Network for Applied Technology
Edmonton, Canada
eden@nat.ltd

## Abstract

The "wisdom of crowds" phenomenon shows that aggregating independent estimates can yield more accurate predictions than individual guesses. While crowd-sourcing is widely applied, using large language models (LLMs) for collective estimation is largely unexplored. This work investigates how to best form an LLM "crowd" for ambiguous vision-based estimation tasks. We explore two sources of diversity: response diversity, from sampling at various temperatures, and model diversity, from using different LLM architectures. We evaluate these approaches on three vision-based datasets: human height-weight pairs, small objects with known weights, and Amazon products with their prices. Our results show that aggregating deterministic (temperature 0) outputs from a diverse set of models is the most effective strategy, outperforming any single model and ensembles that rely on stochasticity from higher temperatures. We find that temperature-induced diversity introduces more noise than signal. The median aggregation of deterministic responses from multiple models outperformed 67% of individual guesses on average, a figure that rises to 75% when relevant context is provided, demonstrating that model diversity is the key to leveraging the wisdom of LLM crowds. By establishing core principles for forming an effective LLM crowd, this work provides a stepping stone for more complex, LLM-driven social simulations.

## 1 Introduction

A century after Galton's famous demonstration of a crowd correctly guessing an ox's weight, the "wisdom of crowds" continues to inspire research in social science and machine learning (Galton, 1907; Surowiecki, 2004). Large language models (LLMs) offer a modern take on this concept. These models are not single, fixed predictors; they generate varied outputs through stochastic processes, each with its own biases, offering different perspectives on a problem (Wei et al., 2022). This response diversity allows treating each model call as an independent agent, like an individual in a crowd. In this view, LLM calls can act as independent computational agents in a synthetic society, offering uninfluenced judgments and enabling simulations of decentralized social systems.

Modern LLMs can produce multiple outputs in one inference call via sampling methods. These outputs represent diverse predictions, each with a numerical estimate and token probabilities that reflect the model's internal uncertainty (Wei et al., 2022). Aggregating outputs from different models, or generating multiple outputs from a single model using various temperatures, lets us explore two distinct sources of diversity. An ensemble of different models provides *model diversity*, where each agent has a unique architecture and training background. Alternatively, varying the temperature for a single model generates *response diversity* from the stochasticity of the sampling process. This is analogous to traditional machine learning ensembles, where combining predictions from different models or samples reduces individual errors and improves overall performance (Hansen & Salamon, 2002). However, unlike classical ensemble methods where base learners are designed to be complementary (Hansen & Salamon, 2002), LLMs produce unstructured, stochastic outputs, making it unclear how best to combine them. Moreover, the relationship between individual sampling variability and collective accuracy in generative language models remains underexplored (Lau et al., 2024).

In this work, we ask: *What is the most effective way to generate and aggregate diverse outputs from LLMs to achieve higher predictive accuracy?* We address this by formalizing LLM aggregation as a computational extension of crowd wisdom and comparing model diversity versus response diversity. Our findings indicate that aggregating deterministic (temperature 0) outputs from a diverse model ensemble yields the most accurate and reliable estimates. This suggests that variability from higher temperatures introduces more noise than signal, and that the most effective "crowd" is a group of diverse "experts," each providing their single most confident answer. Our work suggests that combining outputs from different models can yield reliable numerical estimates without needing larger or more complex models.

We evaluated our approach on three vision-based estimation tasks using datasets for human height and weight, object mass from images, and product prices from Amazon listings. These datasets allow us to focus on predicting numerical values from images, sometimes with additional context. Section 3.1 provides more details on our datasets and methodology.

Surowiecki (2004) defines several criteria for crowd wisdom. In Section 3.1, we discuss these criteria and how our experiments are designed to meet them. We used several vision-enabled LLMs, including Qwen2 Vision Language 72B Instruct and Llama 3.2 Vision 11B (from the Together API[1]), and GPT-4o-mini[2]. To investigate response diversity, we varied the temperature across five settings (0.2, 0.4, 0.6, 0.8, and 1.0) in our initial experiments, treating each API call as an independent agent. This mirrors the conditions of Galton's experiment. By combining these independent guesses, we aimed to find the best strategy for a collective prediction that is more accurate than any single response.

Our results show that the median aggregation of deterministic outputs from diverse models outperforms individual estimates by a significant margin. On average, the aggregate outperforms 67% of individual responses, which improves to 75% when relevant context is provided. We also explore different aggregation methods and the impact of additional context, discussing their relative performance and implications.

## 2 Related Work

Combining independent judgments improves prediction accuracy, a principle recognized in social science and machine learning. In the early 20th century, Galton's ox-weight guessing experiment (Galton, 1907) showed that the median of diverse estimates approximates the true value. This "wisdom of crowds" effect (Surowiecki, 2004) depends on diverse and independent estimates, which help cancel out errors. Simoiu et al. (2019) found that the median outperforms 65% of individual guesses, confirming the effect in humans.

This concept has influenced ensemble methods in machine learning. Techniques like bagging (Breiman, 1996) and boosting (Freund & Schapire, 1996) combine multiple models to

---

[1] https://docs.together.ai/docs/vision-overview
[2] https://openai.com/index/gpt-4o-mini-advancing-cost-efficient-intelligence/

reduce variance and prevent overfitting, often improving accuracy. In computer vision, deep ensembles (e.g., averaging predictions from several deep residual networks) have achieved state-of-the-art performance (He et al., 2016). Similarly, in NLP, aggregating outputs from various language models improves results.

Recent research on large language models (LLMs) shows that some reasoning abilities emerge only in sufficiently large models (Wei et al., 2022). Self-consistency decoding (Wang et al., 2022) combines multiple outputs to improve reliability and accuracy, suggesting that ensemble methods can reveal latent capabilities. Lau et al. (2024) and Guo et al. (2024) explore how varying prompts elicits diverse reasoning outputs. Lau et al. (2024) vary prompt wording to examine problems from different angles, while Guo et al. (2024) use multiple prompts to reduce issues like reasoning hallucinations. Both studies show that careful prompt design enhances the ensemble effect, leading to more reliable predictions.

Pratt et al. (2024) ask whether forecasting strategies can improve LLM decision-making. This research aligns with agent-based models by Gao et al. (2024), where LLMs act as autonomous agents in simulations. Models simulating human behavior, such as in Park et al. (2023), show that artificial agents can mimic social dynamics. Li et al. (2023) examine if LLMs can understand others' beliefs to encourage collaboration, while Shi et al. (2025) propose that agent interactions can reduce reasoning errors.

Schoenegger et al. (2024) compare LLM ensemble forecasts with aggregated human predictions in a forecasting tournament. They used an ensemble of twelve LLMs for binary predictions on 31 questions, comparing the result to 925 human forecasters over three months. Their analysis shows aggregated LLM predictions outperform a no-information benchmark and are statistically indistinguishable from human forecasts (within medium-effect-size equivalence bounds). They also show forecasting accuracy improves when models see the median human prediction, but simply averaging human and machine outputs is best. While their work focuses on binary forecasts, which are a useful benchmark for the wisdom-of-the-crowd effect, our study extends this paradigm to continuous estimation tasks using vision-enabled LLMs. Numerical guesses, like weight or cost, have more direct real-world applicability than binary predictions. By aggregating continuous outputs from LLMs at different temperatures, our approach attempts to utilize prediction variability for more accurate and robust estimates.

This work aligns with the findings of Schoenegger et al. (2024) by demonstrating that ensemble methods are also effective for complex tasks with visual input and continuous-valued outputs. While ensemble learning, prompt diversity, and agent-based modeling are often studied independently, our approach combines all three, treating LLM configurations as heterogeneous agents whose collective output can be systematically combined.

The next section describes our experimental methodology for simulating this ensemble behavior and evaluating its predictive accuracy across different datasets.

## 3 Methodology

### 3.1 Experimental Setup

#### 3.1.1 Datasets and Data Selection

We used three datasets, randomly sampling 100 items from each with a fixed seed (42) for reproducibility. First, from Kaggle's "Height-Weight Images" dataset [3], we used photos of people with their known weight (lbs) and height (feet, inches). Second, from the Image2Mass dataset (Standley et al., 2017), we used photos of small objects with their weight (lbs, converted to grams) and dimensions (inches). Third, from an Amazon Canada listings dataset by Asaniczka[4], we used product images and their prices (CAD).

---

[3]https://www.kaggle.com/datasets/virenbr11/height-weight-images
[4]https://www.kaggle.com/datasets/asaniczka/amazon-canada-products-2023-2-1m-products

| Dataset | Without Context | With Context |
|---|---|---|
| Height-Weight | "Based solely on the image, give your best numeric estimate of the weight (in lbs) of the person. Output only the number and nothing else." | "Based on the image and the additional information that this person is [HEIGHT] tall, give your best numeric estimate of their weight (in lbs). Output only the number and nothing else." |
| Image2Mass | "Based solely on the image, give your best numeric estimate of the weight (in grams) of the object. Output only the number and nothing else." | "Based on the image and knowing that the object has dimensions [DIMENSIONS] inches, give your best numeric estimate of the weight (in grams) of the object. Output only the number and nothing else." |
| Amazon Price | "Based solely on the image, give your best numeric estimate of the price (in CAD) of the product. Output only the number and nothing else." | "Based on the image and the product title '[TITLE]', give your best numeric estimate of its price (in CAD). Output only the number and nothing else." |

Table 1: Prompt templates used for each dataset and context condition. Variables in brackets were replaced with actual values during experiments.

### 3.1.2 Models and Configuration

We used three vision-language models (using the available model versions in June 2025):

- via OpenAI API: **GPT-4o-mini**
- via Together API: **Qwen2-VL-72B-Instruct** and **Llama-3.2-11B-Vision-Instruct-Turbo**

### 3.1.3 Prompts Used

Table 1 provides the exact prompts used in our experiments.

### 3.1.4 Experimental Parameters

For the initial two datasets, we tested five temperature settings for each of the three models: 0.2, 0.4, 0.6, 0.8, and 1.0. Each configuration was repeated 15 times per image, resulting in 225 total API calls per image (3 models × 5 temperatures × 15 repetitions). Based on our finding that temperature adds more noise than signal (see Section 4), our Amazon Price experiments used only temperature 0. For this dataset, each model was queried once per image. All other parameters were held constant: max_tokens=10 and top_p=1.0.

### 3.1.5 Task Definition and Wisdom of Crowds Criteria

The task was to estimate a numerical value (weight or price) from an image.

The "wisdom of crowds" relies on several criteria for a group to produce accurate collective judgments. As identified by Surowiecki (2004), these are:

1. **Diversity:** Each individual contributes unique insights. This variance in opinion helps to counterbalance errors and biases, improving collective accuracy.
2. **Independence:** Judgments must be independent. Uncorrelated errors tend to cancel out when aggregated, making the collective estimate more accurate.
3. **Decentralization:** Decision-making should be decentralized, allowing individuals to use their own knowledge.

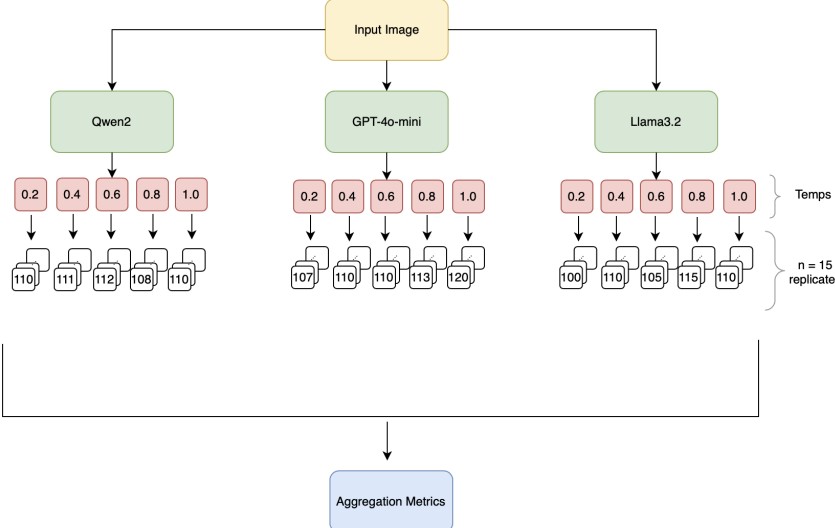

Figure 1: Architecture of the initial LLM ensemble used for the Height-Weight and Image2Mass datasets. The Amazon Price experiments followed a simpler approach.

4. **Aggregation Mechanism:** A mechanism is needed to aggregate individual judgments into a collective decision, from simple averaging to more complex weighted combinations.

These conditions allow for a robust and accurate collective decision that can exceed individual capabilities (Surowiecki, 2004), mirroring principles of decentralized social systems.

### 3.1.6 Experimental Design

To meet these criteria, we experimented with different LLMs in various settings (Figure 1). This allowed us to compare two primary sources of diversity: **model diversity**, from using an ensemble of different models, and **response diversity**, from generating multiple outputs from a single model at various temperatures. Temperature controls output randomness (lower is more deterministic). This will allow us to compare whether a crowd of diverse models is wiser than a crowd of diverse responses from a single model. We treated each API call as an independent agent with no shared context or communication between calls, in an attempt to mimic collecting diverse, independent, and decentralized guesses from a crowd. In Section 3.2, we consider different aggregation methods to satisfy the fourth criterion.

Finally, we aggregated the outputs to see if the collective estimate could outperform individual predictions. Comparing these aggregates to the ground truth allowed us to assess the collective and predictive potential of LLM ensembles. Human crowds often use context to improve their predictions. We simulated this by giving LLMs extra context, hypothesizing it would improve accuracy. For the Height-Weight and Image2Mass datasets, we provided context by including the person's height or the object's dimensions in the prompts. We then compared predictions with and without context to assess their impact. For the Amazon Prices dataset, we gave the product title in the prompt.

## 3.2 Aggregation and Weighting Methods

We combined independent outputs from multiple LLM calls for a robust aggregate estimate. We considered two main unweighted aggregation methods:

1. **Mean:** The arithmetic mean $\bar{x} = \frac{1}{n} \sum_{i=1}^{n} x_i$ (sensitive to outliers).
2. **Median:** The median, the middle value of sorted predictions (robust to outliers).

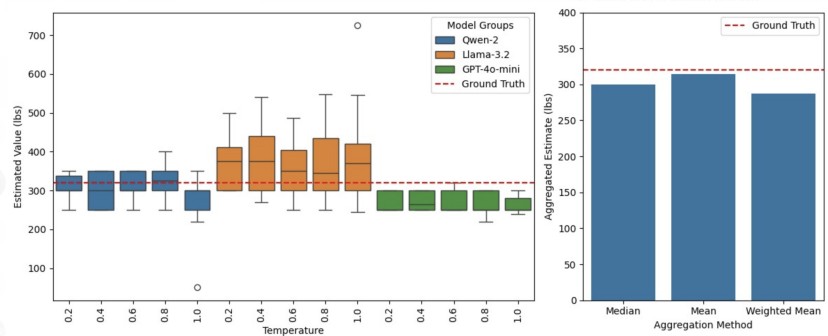

Figure 2: An example from the Height-Weight dataset showing different model biases. Qwen-2 and GPT-4o-mini tend to underestimate weight, while Llama-3.2 overestimates. The median of the estimates is closer to the true value than any single model's.

We also explored weighting predictions by token-level confidence (log-probabilities), but our initial analysis showed no significant benefit over unweighted methods. We therefore focus on the mean and median in our main analysis. All aggregation methods are blind to the ground truth, combining only the LLM outputs.

### 3.3 Ranking

We assess performance using a ranking mechanism similar to Simoiu et al. (2019). For each image, we rank individual prediction errors to find the rank percentile of the aggregated estimate. This percentile shows the fraction of individual predictions the aggregate outperforms (a lower percentile means fewer individual predictions were better). For statistical significance, we use a one-sided paired t-test on the rank percentiles for each image, as we compare methods on the same items.

### 3.4 Mathematical Model

Consider an entity $E$ with a $d$-dimensional attribute vector $\boldsymbol{\theta} \in \mathbb{R}^d$. A digital image of it, $P(E)$, is shown independently to $m$ LLM agents at temperature $T = 0$.

Each agent $A_i$ produces an estimate

$$\hat{\boldsymbol{\theta}}_i = f_i\big(P(E)\big) = \boldsymbol{\theta} + \boldsymbol{\varepsilon}_i, \quad i = 1, \ldots, m, \tag{1}$$

where $\boldsymbol{\varepsilon}_i$ denotes the idiosyncratic error of agent $i$.

We then form a coordinate-wise median aggregation:

$$\tilde{\theta}_j = \mathrm{median}\big\{\hat{\theta}_{1j}, \hat{\theta}_{2j}, \ldots, \hat{\theta}_{mj}\big\}, \quad j = 1, \ldots, d. \tag{2}$$

We assume each $f_i$ is a distinct world model (due to different architectures, data, and biases), so their error vectors $\boldsymbol{\varepsilon}_i$ have zero median and are weakly correlated ($\mathrm{Cov}(\boldsymbol{\varepsilon}_i, \boldsymbol{\varepsilon}_j) \approx 0$ for $i \neq j$). This assumption best applies when aggregating across diverse model architectures, which our results show is most effective.

## 4 Results and Discussion

Our findings show that: 1) median aggregation is more effective than the mean; 2) model diversity is the main driver of performance; 3) temperature-induced diversity adds more noise than signal; and 4) aggregation produces a "wise" crowd whose estimate is significantly better than a typical individual guess. We will now go over each of these statements.

**Median aggregation is most robust.** Our first analysis confirmed that the median is superior to the mean for aggregation. Estimation tasks are prone to outliers, causing the mean to

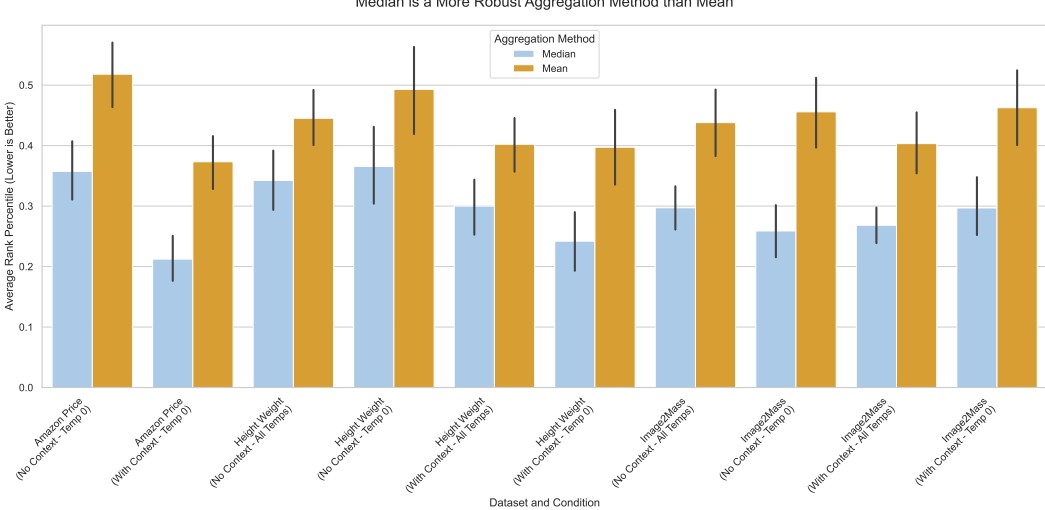

Figure 3: Median vs. Mean aggregation performance. The median consistently achieves a lower (better) average rank percentile, confirming its robustness.

perform poorly as an aggregation method. Our experiments (3) show that the median's rank percentile was significantly lower (better) than the mean's across all datasets and conditions (p < 0.001 for all testable comparisons). We therefore focus on median-based approaches.

**Model diversity drives performance.** Our central hypothesis, that a crowd of diverse models outperforms any single model, is strongly supported by our results. The "All Models" median aggregate consistently achieved a better rank percentile than any single model's median, as shown in Figure 4. For instance, the "All Models" median at Temperature 0 was significantly better than 'GPT4o-mini' alone (p = 0.0017, height-weight) and was never significantly worse than any single model. This confirms that aggregating estimates from diverse models cancels individual biases (Figure 2).

**Temperature adds noise, not signal.** We questioned if stochasticity from higher temperatures provides useful diversity. It does not (Figure 4). A paired t-test reveals no significant performance difference between 'Temp 0' and 'All Temps' median aggregates, which shows temperature-induced randomness is not a source of "wisdom," but adds statistical noise (and additional API calls) without improving the estimate. Thus, the most effective strategy is to use the single most confident guess (Temperature 0) from each diverse model.

**Context benefit is task-dependent.** Providing context (e.g., height) significantly improved performance for the 'height-weight' dataset (p < 0.01). Similarly, for the Amazon Price dataset, context (product title) significantly improved accuracy (p < 0.0001). However, for 'image2mass', context (dimensions) had no significant effect. This suggests context is beneficial only if relevant to the model's reasoning for that task.

**Aggregates are better than individuals.** To test for a "wise" crowd, we checked if our best method's rank percentile ('Median - All Models - Temp 0') was significantly below 0.5 (the expected rank of an average individual). A one-sample t-test confirms this with high confidence (p < 0.0001) across all conditions. Table 2 and Figure 5 summarize this result. This confirms that the aggregate is not just better than a few noisy individuals but is statistically superior to the typical individual guess, which shows a wisdom of the crowd effect. On average, our method outperforms 67% of individual responses without context, rising to 75% with context.

| Dataset | Without Context | With Context |
|---|---|---|
| Height-Weight | 0.366 | 0.242 |
| Image2Mass | 0.259 | 0.297 |
| Amazon Price | 0.357 | 0.212 |
| **Average** | **0.327** | **0.250** |

Table 2: Average rank percentile of the best aggregate method (Median of All Models, Temp 0). In all conditions, the aggregate significantly outperforms the expected rank of an average individual (0.5), with p < 0.0001 for a one-sample t-test.

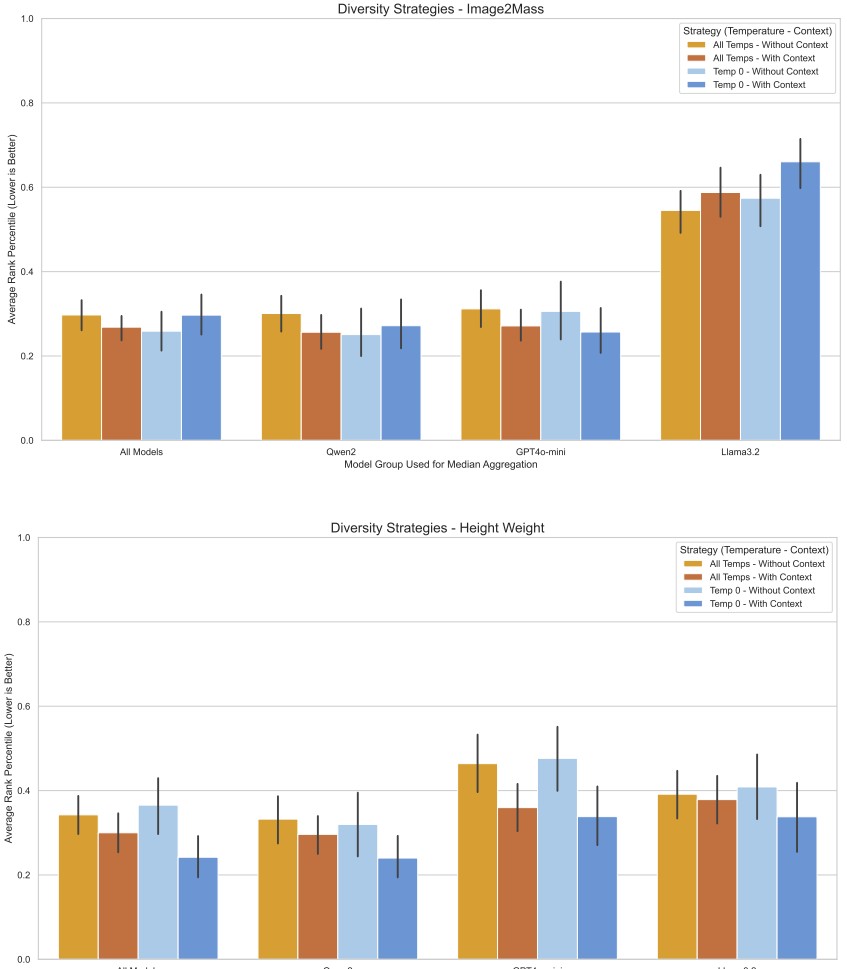

Figure 4: **Diversity strategies.** Median aggregation performance across model groups and strategies. Aggregating "All Models" (leftmost) performs well, and Temperature 0 (blue) is comparable to or better than All Temperatures (orange). This plot shows results for the two datasets where temperature was varied; the Amazon Price dataset, which only used Temperature 0, is omitted as it does not have an "All Temps" condition to compare against.

## 5 Limitations and Future Work

Our study has several limitations that suggest directions for future research. First, our analysis uses three datasets with small sample sizes (100 instances each). Larger, more diverse datasets are needed to generalize our conclusions. Second, our experiments only

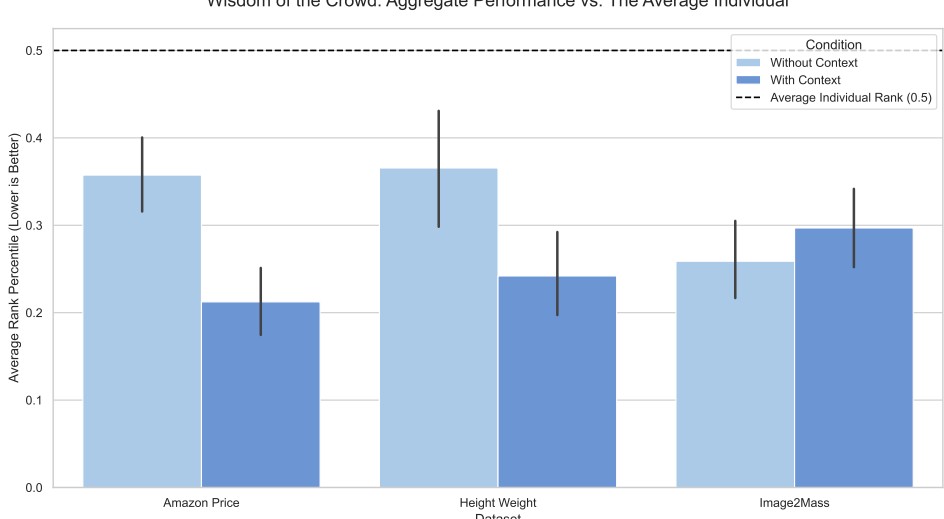

Figure 5: **The aggregate is wiser than the average individual.** Performance of our best method (Median of All Models, Temp 0). Bars show the average rank percentile (95% CI). The dashed line at 0.5 is the expected rank of an average individual. The aggregate is significantly better in all cases.

cover estimating weight and price. As Simoiu et al. (2019) noted, crowd performance varies by task. Future work should test our findings on other tasks, like forecasting or subjective judgments. Third, we lack a direct comparison to a human baseline for these visual tasks, which would provide valuable context.

In future work, we will investigate how sampling parameters (e.g., top-$p$, top-$k$) affect output diversity and aggregate accuracy. We also aim to use token-level distributions (entropy, perplexity) for more refined aggregation weighting. We will explore dynamic multi-agent interactions where agents adjust predictions based on cues and peer outputs, simulating social learning. This would let us model influence and belief propagation as in human social networks, enabling large-scale experiments on collective intelligence. Such simulations could systematically test variables like communication topology and information cascades at a scale and with a level of control that is infeasible in human studies. Finally, we plan to use a Fermi-inspired estimation[5] strategy, using chain-of-thought prompting to make LLMs decompose complex tasks into components (e.g., material, dimensions). Estimating and combining these components may enhance final prediction accuracy.

## 6 Conclusion

Our work shows that "wisdom of the crowds" principles apply to LLM ensembles for vision-based estimation. We find that the source of diversity is critical, as our results demonstrate that aggregating deterministic (temperature 0) outputs from diverse models (model diversity) is more effective than generating multiple outputs from a single model using temperature sampling (response diversity). This is supported by two of our main findings: 1) aggregating across models consistently outperformed any single model, and 2) temperature-induced randomness added noise without improving estimates. Therefore, the most robust strategy is a committee of diverse "expert" models, each giving its most confident estimate. This work provides empirical support for LLMs in agent-based modeling and shows that the path to collective intelligence is model diversity, not sampling randomness.

---

[5] https://en.wikipedia.org/wiki/Fermi_problem

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
