# OpenReview forum: "Wisdom of the Machines: Exploring Collective Intelligence in LLM Crowds"
_colmweb.org/COLM/2025/Workshop/Social_Sim — Social Sim'25_

### Official Review · Reviewer_Xajg · 2025-07-17
**Wisdom of the Machines: Exploring Collective Intelligence in LLM Crowds**

**Rating:** 8
**Overall Assessment:** 4
**Confidence:** 5

**Review:**

The paper is clearly written and easy to read, and covers a very interesting and unexplored topic. The results this paper presents are presented well and follow the original research question, though some of the assumptions this paper makes seem questionable

# Pros
- Well written
- Results follow the narrative that the research question presents
- Novel topic
- Acknowledges most experimental limitations well.

# Cons
- Questionable assumptions made relating to the experiment's adherence to Surowiecki's criteria for crowd wisdom
- Limited experimental settings and lack of exploration

**Comments Suggestions And Typos:**

1. Moving the last sentence on line 83 to the start of the following paragraph would make it more clear
2. Increase the font size of Figure 3 for readability. You can also make the angle a little straighter.
3. Would love to see if there is a relationship between the number of API calls and/or model architectures and median performance( larger crowd = more likely to be correct?).


See weaknesses section for more content/exploration suggestions

**Paper Summary:**

In this paper, the authors come up with a framework to evaluate the utility of using numerous vision LLM responses to respond to questions with numeric answers involving visual estimation of certain attributes of some subject in the image. The authors find that using a larger number of vLLM architectures leads to better performance compared to varying sampling temperature. They also find that using the median of LLM responses provides a better estimate of all the responses compared to using the mean. Finally, the authors show that crowd vLLM responses outperform most individual vLLM responses, and further outperforms them when given context to use to solve the problem.

**Relevance:**

5

**Summary Of Strengths:**

Aside from writing, the main strength of this paper is that the results directly support an answer to the research question: "What is the most effective way to generate and aggregate diverse outputs from LLMs to achieve higher predictive accuracy". The authors come up with an experimental design that's based in literature and findings in other related fields. Further, they then evaluate two different metrics for aggregating outputs from various vLLM responses.

There is limited work in evaluating the crowd wisdom effects in LLMs, and this seems to be the first that evaluates them on vision-based numeric-answer tasks.

**Summary Of Weaknesses:**

The main weakness of this study is in the assumptions it makes about the first aspect detailed in Section 3.1.5, diversity. It is not necessarily straight forward that different LLM architectures would provide variance in opinion, and even less straight forward to assume different API calls could safely act as different agents. Many LLMs are trained on similar types of data, what would lead you to believe that they provide "diverse" opinions? Given the same three models, does 225 API calls across them really provide a diverse set of perspectives? I believe a discussion of this is needed at the very least. It could also be an interesting direction to try and quantify the diversity in LLM responses as well.

Relatedly, only using three LLM architectures is another issue that isn't addressed in the limitations section. Assuming LLM architectures provide diverse perspectives from one another, then adding more LLM architectures could bring the mean/median down and change the nature of your results.

---

### Meta-Review · Area_Chair_D2Zo · 2025-07-21

**Recommendation:** Accept

**Metareview:**

--